# OPG/TRAIL ratio as a predictive biomarker of mortality in patients with type A acute aortic dissection

Jie Lu [1,8], Ping Li[1,8], Ke Ma[1], Yang Li[1], Hui Yuan[1], Junming Zhu[1], Weixun Duan[2], Jingsong Ou[3], Yonghong Huang[4], Long Wu[5], Xueliang Pan [6], Hui Zhang[7], Jie Du [1✉] & Yulin Li [1✉]

Following hospital discharge, patients with type A acute aortic dissection (TA-AAD) may present an increase in mortality risk. However, little is known about specific biomarkers associated with post-discharge survival, and there is a paucity of prognostic markers associated with TA-AAD. Here, we identify nine candidate proteins specific for patietns with TA-AAD in a cross-sectional dataset by unbiased protein screening and in-depth bioinformatic analyses. In addition, we explore their association with short-term and long-term mortality in a derivation cohort of patients with TA-AAD, including an internal ($n = 300$) and external ($n = 236$) dataset. An elevated osteoprotegerin (OPG)/tumour necrosis factor-related apoptosis-inducing ligand (TRAIL) ratio was the strongest predictor of overall, 30-day, post-30-day mortality in both datasets and was confirmed to be a strong predictor of mortality in an independent validation cohort ($n = 400$). Based on OPG/TRAIL ratio-guided risk stratification, patients at high risk (>33) had a higher 1-year mortality (55.6% vs. 4.3%; 68.2% vs. 2.6%) than patients at low risk (<4) in both cohorts. In Conclusion, we show that an elevated OPG/TRAIL ratio is associated with a significant increase in short-term and long-term mortality in patients with TA-AAD.

[1] Beijing Anzhen Hospital of Capital Medical University and Beijing Institute of Heart Lung and Blood Vessel Diseases, Beijing, China. [2] Xi Jing Hospital of Fourth Military Medical University, Xian, China. [3] First Affiliated Hospital of Sun yat-sen university, Guangzhou, China. [4] First hospital affiliated to Dalian medical university, Dalian, China. [5] Union Hospital Affiliated with Tongji Medical College, Huazhong University of Science and Technology, Wuhan, China. [6] Center for Biostatistics, the Ohio State University, Columbus, USA. [7] Department of Preventive Medicine, Feinberg School of Medicine, Northwestern University, Chicago, USA. [8] These authors contributed equally: Jie Lu, Ping Li. ✉email: jiedu@ccmu.edu.cn; lyllyl_1111@163.com

A ortic dissection is now encountered more often because of the increasing prevalence of arterial hypertension in the population. Type A acute aortic dissention (TA-AAD) is a catastrophic condition caused by dissection of the ascending aorta[1]. Despite many therapeutic improvements, in particular timely and successful surgery, in-hospital mortality rates still approach 22%[2]. Moreover, TA-AAD is a progressive condition involving the entire aorta and its branches, which remain at high risk even after treatment of the acute index event. The clinical outcomes of TA-AAD vary widely[3]. Accurate risk stratification influences further treatment decisions (e.g., choice of type of surgery)[4] and management post-discharge (e.g., serial radiological monitoring and medical therapy)[5,6]. Timely identification of patients at high risk will help improve the overall prognosis of TA-AAD. Therefore, accurate risk stratification of patients with TA-AAD is an urgent task.

There have been several prognostic studies of short-term (in-hospital or 30-day) mortality in TA-AAD. Some studies, including post-hoc analyses from the International Registry of Acute Aortic Dissection (IRAD) or the German Registry for AAD, identified several predictors of worse outcomes in patients with TA-AAD[7,8], but without proposing a risk stratification model. Although risk scores based on preoperative or operative variables have been developed[9,10], they have not been validated. The ability to predict the long-term risk of death in patients with TA-AAD is important for two reasons. First, improving surgical techniques and intensive care facilities has increased the overall life expectancy of patients with TA-AAD. Post-discharge mortality rate was reported to be 4% at 1 year and to increase to 10% at 3 years in patients with TA-AAD treated by surgery[3]. Thus, although life expectancy has increased, survivors consistently remain at greater risk of post-discharge mortality. Second, although strategies to reduce acute mortality are well recognised, no systematic efforts are in place to reduce the risk of death that occurs after hospital discharge. Few studies have described independent predictors of mortality during long-term follow-up of patients with TA-AAD. Initial attempts to create prognostic algorithms for risk stratification based on clinical data alone have been unsuccessful, and there is an ongoing need to develop a biomarker-based prognostic tool. Biomarkers could play an essential role as objective tools for diagnosis and prognosis. The pathological processes leading to TA-AAD cause characteristic changes in signalling proteins in the circulation, generating detectable disease-specific molecular biomarkers. Although several circulating proteins, including D-dimer[11,12], serum potassium[13], and creatinine[14], may predict adverse outcomes in patients with TA-AAD, almost all of these proteins share the limitations of not having been validated, being derived from small studies, and have only weak associations with long-term outcomes.

In this work, we attempt to identify a biomarker associated with short-term and long-term mortality in patients with TA-AAD. Here, we screen nine TA-AAD-related proteins and verify the ratio of osteoprotegerin (OPG)/tumour necrosis factor-related apoptosis-inducing ligand (TRAIL) as the strongest marker for prediction of the mortality in two independent cohorts. A risk stratification based on OPG/TRAIL ratio is developed and well-validated. OPG/TRAIL ratio at admission is a robust prognostic marker for both short-term and long-term death in patients with TA-AAD.

## Results

### Characteristics of the study population in the screening and derivation stage.
Clinicopathological characteristics of TA-AAD patients ($n = 120$) and healthy controls (HCs) ($n = 244$) in the screening sets are summarized (Supplementary Table 1). A total of 536 patients in the derivation cohort (including 300 in the internal dataset and 236 in the external dataset) were included. The baseline clinical characteristics of the patients are summarized in Table 1 according to cohort and stratified by death status. Forty-one patients (13.7%) in the internal dataset and 50 patients (21.2%) in the external dataset died. Patients in the internal dataset who died were more likely to have coma/stroke, a lower diastolic blood pressure, a lower body mass index (BMI), and less hypertension, while those in the external dataset were more likely to have higher body mass index and be older, and have less history of aortic aneurysm. In both datasets, surviving patients were more likely to undergo surgical treatment than those who died.

### Exploration of candidate proteins related to TA-AAD.
The primary aim of this study was to identify candidate proteins closely related to TA-AAD. In the first screening step, a commercialised antibody array containing 1000 proteins showed a significant difference in 41 proteins between the two groups after the initial statistical analyses ($p < 0.05$, Supplementary Table 2, Supplementary Data 1). Twenty-six proteins were selected on the basis of their relationship with the pathology of aortic dissection by using two algorithms (see Supplementary Methods and Supplementary Fig. 1a for details of the selection strategies); the top-ranking candidate protein was D-dimer (a known biomarker of aortic dissection). In the second screening step, serum levels of 26 proteins from 31 patients with TA-AAD and 32 healthy controls were measured using a custom antibody array, and a significant difference in 14 of the 26 proteins remained between the two groups [fold change (FC) > 1.5; $p < 0.05$] (Supplementary Fig. 1b). In the third screening step, the serum concentrations of these 14 proteins were quantified in 77 patients with TA-AAD and 200 healthy controls by enzyme-linked immunosorbent assay. There were significant difference in nine of these proteins, namely, serum amyloid A 1 (SAA1), plasminogen (PLG), angiopoietin-1 (ANGPT1), platelet factor 4 (PF4), osteoprotegerin (OPG), fibronectin (FN1), tumour necrosis factor (TNF)-related apoptosis-inducing ligand (TRAIL), oxidised low-density lipoprotein receptor 1 (LOX-1), and lipocalin 2 (LCN2) between the patients with TA-AAD and the healthy controls (Supplementary Fig. 1c–o), suggesting that they may be TA-AAD-related proteins.

### Association of the OPG/TRAIL ratio with risk of death in the derivation cohort.
The secondary aim of this study was to identify a robust prognostic biomarker associated with the risk of death in patients with TA-AAD. The serum concentrations of the nine above-mentioned candidate proteins were tested in blood samples collected at admission in patients from both the internal and external datasets. Because OPG binds specifically to TRAIL and neutralises its function, we also calculated the OPG/TRAIL ratio based on OPG and TRAIL concentrations. In both datasets, the patients who died had significantly higher concentrations of OPG and lower concentrations of TRAIL than patients who survived in both cohorts (Supplementary Fig. 2); thus, the OPG/TRAIL ratio in patients who died was higher than in patients who survived in both the internal dataset and external dataset.

We analysed the predictive/discriminative value of the nine candidate biomarkers, the OPG/TRAIL ratio and D-dimer. First, Cox regression analysis showed that the OPG/TRAIL ratio was the strongest independent predictor of overall mortality (adjusted hazard ratio [HR] 2.04 [95% confidence interval [CI] 1.48–2.82]; 2.64 [95% CI 1.71–4.07]), 30-day mortality (adjusted HR 2.05 [95% CI 1.35–3.12]; 2.33 [95% CI 1.42–3.82]) and post-30-day mortality (adjusted HR 2.07 [95% CI 1.25–3.43]; 4.68 [95% CI 1.72–12.69]) in the internal and external datasets, respectively (Fig. 1, Supplementary Table 3). Higher D-dimer concentrations were associated with increased 30-day mortality but was not with overall mortality, which is consistent with previous reports[11,12]. Second,

**Table 1 Patient characteristics in the derivation and validation cohorts.**

| | Derivation cohort | | | | | | Validation cohort | | |
| | Internal set | | | External set | | | | | |
| Characteristics Demographics | Death (n = 59) | Survival (n = 241) | p-value | Death (n = 50) | Survival (n = 186) | p-value | Death (n = 88) | Survival (n = 312) | p-value |
|---|---|---|---|---|---|---|---|---|---|
| Age (y) | 50.1 ± 13.1 | 47.5 ± 11.0 | 0.110 | 53.5 ± 12.5 | 48.6 ± 11.8 | 0.012 | 57.2 ± 13.2 | 48.7 ± 11.3 | 5.000e-6 |
| Male | 43 (72.9) | 198 (82.2) | 0.108 | 35 (70.0) | 146 (78.5) | 0.207 | 58 (65.9) | 245 (78.5) | 0.015 |
| Female | 16 (27.1) | 43 (17.8) | 0.108 | 15 (30.0) | 40 (21.5) | 0.270 | 22 (34.1) | 67 (21.5) | 0.015 |
| BMI (kg/m$^2$) | 24.2 ± 4.2 | 26.6 ± 3.9 | 3.260e-4 | 27.5 ± 4.1 | 25.8 ± 3.3 | 0.010 | 25.0 ± 4.3 | 26.7 ± 4.5 | 0.015 |
| Smoking | 24 (40.7) | 105 (43.6) | 0.688 | 17 (34.0) | 70 (37.6) | 0.636 | 30 (34.1) | 134 (43.0) | 0.136 |
| SBP (mmHg) | 134.2 ± 28.7 | 131.7 ± 21.4 | 0.540 | 137.8 ± 31.2 | 134.3 ± 20.3 | 0.451 | 129.4 ± 23.3 | 129.1 ± 21.9 | 0.916 |
| DBP (mmHg) | 68.3 ± 16.6 | 74.3 ± 14.4 | 0.007 | 75.7 ± 20.2 | 74.3 ± 14.7 | 0.573 | 73.5 ± 14.0 | 72.0 ± 14.9 | 0.388 |
| Patient history | | | | | | | | | |
| Hypertension | 31 (52.5) | 168 (69.7) | 0.012 | 36 (72.0) | 132 (71.0) | 0.886 | 68 (77.3) | 256 (82.1) | 0.313 |
| Hyperlipemia | 1 (1.7) | 10 (4.1) | 0.698 | 2 (4.0) | 6 (3.2) | 0.678 | 8 (9.1) | 38 (12.2) | 0.422 |
| DM | 6 (10.2) | 18 (7.5) | 0.591 | 4 (8.0) | 14 (7.5%) | 1.000 | 9 (10.2) | 18 (5.8) | 0.141 |
| CAD | 1 (1.7) | 11 (4.6) | 0.472 | 4 (8.0) | 9 (4.8) | 0.482 | 15 (17.1) | 32 (10.3) | 0.081 |
| Marfan syndrome | 3 (5.1) | 6 (2.5) | 0.386 | 0 (0) | 3 (1.6) | 1.000 | 0 (0) | 2 (0.6) | 1.000 |
| BAV history | 3 (5.1) | 6 (2.5) | 0.387 | 1 (2.4) | 2 (1.1) | 0.463 | 1 (1.1) | 7 (2.2) | 1.000 |
| Known of AA | 11 (18.6) | 38 (15.8) | 0.592 | 2 (4.0) | 38 (20.4) | 0.006 | 6 (6.8) | 31 (9.9) | 0.373 |
| Presenting Symptoms and signs | | | | | | | | | |
| AVI | 24 (40.7) | 131 (54.8) | 0.052 | 15 (38.5) | 94 (52.5) | 0.112 | 43 (48.9) | 136 (57.4) | 0.156 |
| High-risk pain | 39 (66.1) | 171 (71.0) | 0.466 | 45 (90.0) | 158 (84.9) | 0.360 | 55 (62.5) | 184 (59.0) | 0.551 |
| Acute MI | 2 (3.4) | 21 (8.7) | 0.272 | 9 (18.0) | 23 (12.4) | 0.302 | 8 (9.1) | 17 (5.5) | 0.213 |
| Malperfusion | 12 (20.3) | 51 (21.2) | 0.889 | 13 (26.0) | 60 (32.3) | 0.395 | 34 (38.6) | 88 (28.2) | 0.061 |
| Tamponade | 3 (2.9) | 12 (5.0) | 0.973 | 10 (20.0) | 22 (11.8) | 0.134 | 18 (20.5) | 69 (22.1) | 0.739 |
| Coma or stroke | 5 (8.5) | 6 (2.5) | 0.044 | 5 (10.0) | 8 (4.3) | 0.156 | 5 (5.7) | 27 (8.7) | 0.364 |
| Shock/ hypotension | 3 (5.1) | 7 (2.9) | 0.419 | 3 (6.0) | 6 (3.2) | 0.404 | 8 (9.1) | 7 (2.2) | 0.007 |
| Imaging | | | | | | | | | |
| Thrombosis in false lumen | 22 (37.3) | 99 (41.1) | 0.595 | 18 (36.0) | 78 (41.9) | 0.448 | 24 (27.3) | 136 (43.6) | 0.006 |
| Dissection-involved segments | | | 1.000 | | | 0.425 | | | 0.120 |
| 1 segment | 7 (13.2) | 34 (14.9) | | 4 (11.4) | 14 (8.5) | | 4 (4.6) | 14 (4.6) | |
| 2 segments | 4 (7.5) | 18 (7.9) | | 6 (17.1) | 12 (7.3) | | 9 (10.2) | 24 (7.7) | |
| 3 segments | 9 (17.0) | 37 (16.2) | | 3 (8.6) | 21 (12.7) | | 16 (18.2) | 23 (7.4) | |
| 4 segments | 7 (13.2) | 29 (12.7) | | 5 (14.3) | 25 (15.2) | | 16 (18.2) | 19 (6.1) | |
| 5 segments | 8 (15.2) | 32 (14.0) | | 2 (5.7) | 20 (12.1) | | 18 (20.5) | 34 (10.9) | |
| 6 segments | 18 (34.0) | 78 (34.2) | | 15 (42.9) | 73 (44.2) | | 17 (19.3) | 115 (36.9) | |
| Surgical treatment | 41 (69.5) | 222 (92.1) | 2.000e-6 | 29 (58.0) | 170 (91.4) | 8.109e-9 | 67 (76.1) | 296 (94.9) | 8.441e-8 |

Values are mean ± SD, n (%), or median (interquartile range) unless otherwise indicated. SBP, Systolic blood pressure, DBP, Diastolic blood pressure, DM, Diabetes mellitus, CAD, Coronary artery disease, BAV, Bicuspid aortic valve, AVR, Aortic valve replacement, AVI, Aortic insufficiency, AA, aortic aneurysm, MI, Myocardial infarction. p-values are two-tailed from variance/Kruskal Wallis test for continuous variables or Chi-square test for categorical variables.

the Harrell's concordance index (C-index) for the OPG/TRAIL ratio was the highest for overall death (0.72 [95% CI 0.65–0.78]; 0.77 [95% CI 0.71–0.83]), 30-day death (0.72 [95% CI 0.64–0.81]; 0.75 [95% CI 0.68–0.83]), and post-30-day death (0.72 [95% CI 0.61–0.83]; 0.82 [95% CI 0.71–0.92]) in the internal and external datasets, respectively (Fig. 2, Supplementary Table 4). These data demonstrate that the OPG/TRAIL ratio was the strongest predictor among the nine candidate biomarkers and D-dimer.

**Performance of the OPG/TRAIL ratio compared with clinical variables in the derivation cohort.** We further investigated whether adding the OPG/TRAIL ratio to the clinical variables would improve the risk estimation. We chose the existing AAD score[9] as the initial model for predicting 30-day mortality and chose seven known clinical predictors into the initial models for predicting post-30-day mortality due to the lack of an established risk model. For overall death, the OPG/TRAIL ratio showed fair discrimination (C-index: 0.74 vs. 0.55 for the AAD score and 0.63 for clinical predictors). The OPG/TRAIL ratio increased the C-index significantly when added to the AAD score (ΔC-index 0.19 [95% CI 0.11–0.27]) and clinical predictors (ΔC-index 0.13 [95% CI 0.08–0.18]). Addition of the OPG/TRAIL ratio improved the reclassification of the AAD score (net reclassification index [NRI] 0.56 [95% CI 0.52–0.60]; NRI$_e$ 0.18 [95% CI 0.11–0.26]; NRI$_{ne}$ 0.38 [95% CI 0.33–0.42]) or clinical predictors (NRI 0.23 [95% CI 0.20–0.27]; NRI$_e$ 0.06 [95% CI 0.01–0.10]; NRI$_{ne}$ 0.18 [95% CI 0.14–0.21]; Table 2, Supplementary Fig. 3a, b). There was a similar NRI pattern for the OPG/TRAIL ratio regarding 30-day

and post-30-day deaths in patients with and without events (Supplementary Table 5).

**Development of OPG/TRAIL ratio–guided risk stratification in the derivation cohort.** To enable potential translation into clinical practice, we explored the value of the OPG/TRAIL ratio for risk stratification. Based on the criteria for threshold selection, two cut-off OPG/TRAIL ratio values (4 and 33) were chosen for categorising 95 patients (17.7%) into low-risk (<4) yielding NPV of 95% and 63 patients (11.8%) into high-risk (>33) yielding PPV of 55% (Table 3, Supplementary Fig. 4). Survival analysis showed that patients at high risk had a lower 1-year overall survival rate than those in the low-risk group (44.4% vs. 95.7%), even in the patients that underwent surgery (56.9% vs. 98.8%, Fig. 4).

**Predictive value of OPG/TRAIL ratio in the validation cohort.** The baseline clinical characteristics of the validation cohort are summarised in Table 1. The 30-day and overall mortality rates were 19.0% and 22.0%, respectively. The 30-day mortality was higher in the validation cohort than in the derivation cohort because of the higher rupture rate during hospitalization in the validation cohort (9.0% vs. 6.5%). Patients who died were more likely to be female, older and have shock/hypotension, and lower BMI, a lower thrombosis in a false lumen and a lower surgical treatment rate than those who survived.

We validated the predictive value of the OPG/TRAIL ratio for the risk of death. The OPG/TRAIL ratio was an independent predictor of overall mortality (adjusted HR 3.23 [95% CI 2.12–4.91]), 30-day mortality (adjusted HR 1.84 [95% CI

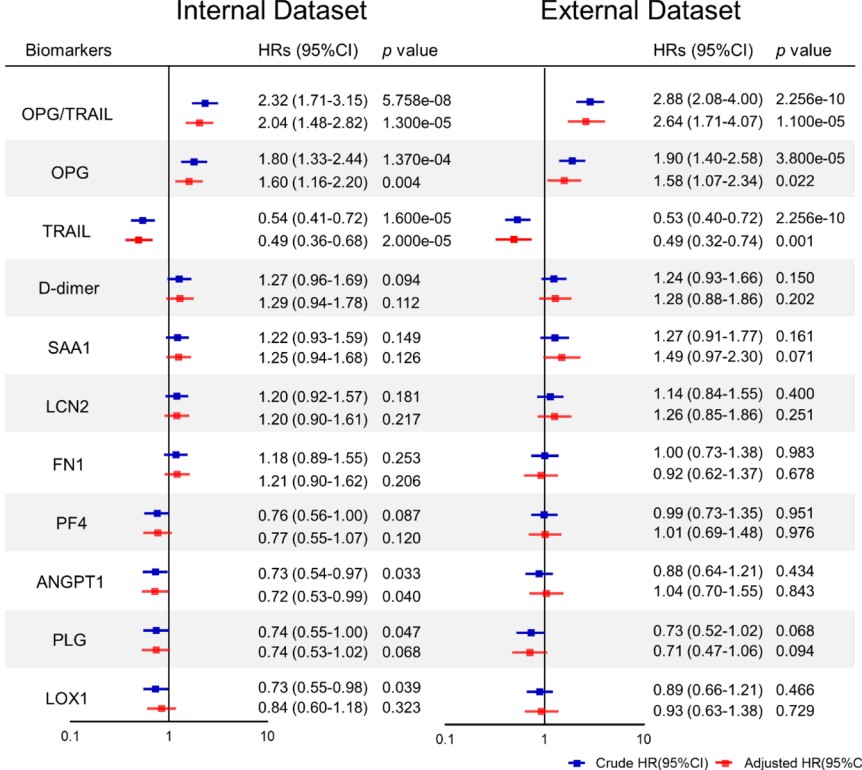

**Fig. 1 Cox proportional hazard analysis of candidate biomarkers for predicting risk of overall death in internal and external datasets of derivation cohort.** Hazard ratios (HRs) with 95% CIs associated with 1-SD increase in 9 protein levels, D-dimer, and OPG/TRAIL ratio for overall death in the internal dataset (n = 300) and external dataset (n = 265) were plotted. Models were adjusted for known risk factors (age ≥ 70 yrs, SBP, smoking), reported potential predictors (pain, malperfusion, shock or hypotension, coma or stroke), imaging indicators (thrombosis in false lumen and segments), and surgical treatment. Boxes represent HR. Error bars represent 95% CI. p values reported are two-tailed from COX proportional hazard regression analyses.

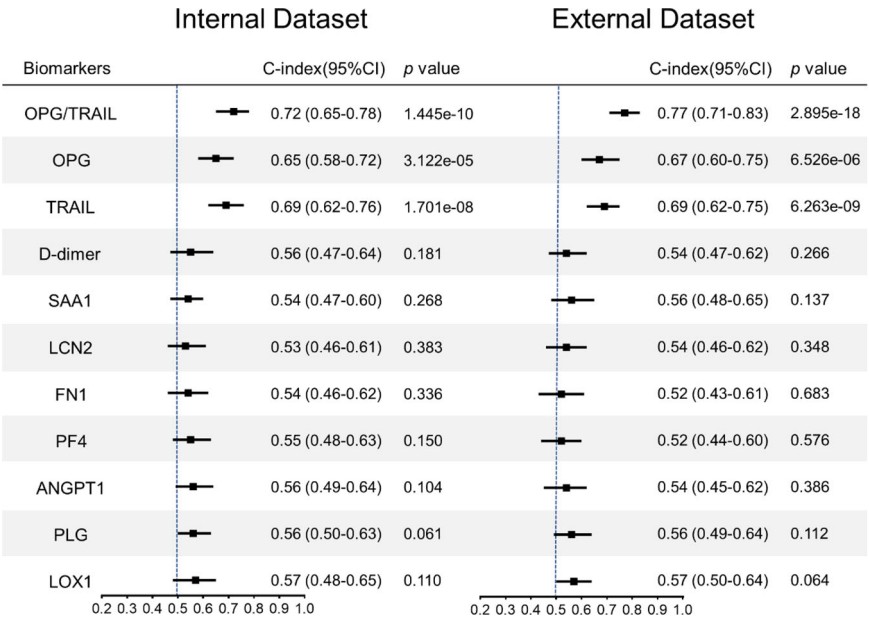

**Fig. 2 C-index of candidate biomarkers for discriminating overall death in internal and external datasets of derivation cohort.** Forest plots showing C-indexes with 95% CIs in the internal dataset (n = 300) and external dataset (n = 265). Boxes represent C-index. Error bars represent 95% CI. p values reported are two-tailed from C-index analyses.

2.82–4.37]), and post-30-day mortality (adjusted HR 8.27 [95% CI 2.19–31.27]; Fig. 3). For overall mortality, The OPG/TRAIL ratio significantly increased the C-index when added to the AAD score (ΔC-index 0.21 [95% CI 0.13–0.29] and clinical predictors

(ΔC-index 0.12 [95% CI 0.07–0.18]). Addition of the OPG/TRAIL ratio improved reclassification of the AAD score (Three-category NRI 0.73 [95% CI 0.68–0.77]; NRI$_e$ 0.32 [95% CI 0.22–0.42]; NRI$_{ne}$ 0.41 [95% CI 0.35–0.46]) or clinical predictors

**Table 2 C-index and three-categories NRI of OPG/TRAIL ratio for predicting the risk of overall death in the derivation and validation cohorts.**

| | Derivation cohort | | | Validation cohort | | |
|---|---|---|---|---|---|---|
| | C-index (95%CI) | ΔC-index (95%CI) | NRI (95%CI) | C-index (95%CI) | ΔC-index (95%CI) | NRI (95%CI) |
| O/T ratio | 0.74 (0.69–0.79) | | | 0.76 (0.71–0.81) | | |
| AAD score | 0.55 (0.48–0.63) | | | 0.55 (0.48–0.62) | | |
| AAD score +O/T ratio | 0.74 (0.70–0.79) | 0.19 (0.11–0.27) | 0.56 (0.52–0.60) Events: 0.18 (0.11–0.26) No-events: 0.38 (0.33–0.42) | 0.76 (0.71–0.81) | 0.21 (0.13–0.29) | 0.73 (0.68–0.77) Events: 0.32 (0.22–0.42) No-events: 0.41 (0.35–0.46) |
| Clinical Predictors | 0.63 (0.58–0.68) | | | 0.66 (0.60–0.72) | | |
| Clinical Predictors +O/T ratio | 0.76 (0.71–0.80) | 0.13 (0.08–0.18) | 0.23 (0.20–0.27) Events: 0.06 (0.01–0.10) No-events: 0.18 (0.14–0.21) | 0.79 (0.74–0.83) | 0.12 (0.07–0.18) | 0.50 (0.45–0.55) Events: 0.19 (0.11–0.28) No-events: 0.31 (0.26–0.36) |

Clinical predictors included stroke, chronic renal dysfunction, myocardial infarction, older age, female, atherosclerosis history, and previous cardiac surgery. O/T = OPG/TRAIL.

(Three-categories NRI 0.50 [95% CI 0.45–0.55]; $NRI_e$ 0.19 [95% CI 0.11–0.28]; $NRI_{ne}$ 0.31 [95% CI 0.26–0.36]; Table 2, Supplementary Fig. 3c, d). The discrimination/reclassification ability of OPG/TRAIL ratio remained to be well for 30-day death and post-30-day death (Supplementary Table 5). In addition, two-category NRI for OPG/TRAIL ratio improved as a result of reclassification of both participants who died or did not die in two cohorts for overall death (Supplementary Table 6).

Next, we evaluated the performance of OPG/TRAIL-based risk stratification in the validation cohort. According to the cut-off values for the OPG/TRAIL ratio derived from the derivation cohort, the cut-off value of <4 had an NPV of 96.2% and a sensitivity of 96.6% whereas the cut-off value >33 had a PPV of 70.5% and a specificity of 95.8% for predicting the overall risk of death. Moreover, the cut-off OPG/TRAIL ratio values also showed a better NPV (97.4%) and PPV (61.4%) for 30-day mortality in the validation cohort (Supplementary Table 7). The PPV value of 23.5% was low for post-30-day mortality due to the low number of deaths ($n = 4$). Survival analyses showed that patients at high risk in the validation cohorts had a lower 1-year overall survival rate than patients at low risk (31.8% vs. 97.4%), even in the patients that underwent surgery (34.3% vs. 97.3%; Fig. 4).

**Linear association and subgroup analyses.** Multivariable-adjusted restricted cubic spline analyses of the association between the OPG/TRAIL ratio and overall mortality provided no evidence of a nonlinear association ($p = 0.244$; $0.231$) and indicated a significant linear association ($p < 0.001$) in the derivation and validation cohorts (Supplementary Fig. 5). Subgroup analysis showed that the HRs for the OPG/TRAIL ratio were comparable across the various predefined subgroups, except for individuals with hypertension in the derivation cohort (Supplementary Fig. 6, Supplementary Table 8). For these individuals, the OPG/TRAIL ratio-associated risk was higher (HR 3.44 [95% CI 2.55–4.66]) than that for individuals without hypertension (HR 1.64 [95% CI 1.16–2.33]).

## Discussion

To date, prognostic biomarker studies in TA-AAD have been limited to known biomarkers and have had a retrospective design[15,16]. Given the low incidence of TA-AAD, our study population comprising 536 patients in the derivation cohort and 400 in the validation cohort, was relatively large. By performing unbiased protein screening and in-depth bioinformatic analyses followed by a multicenter longitudinal study, we have demonstrated that a high OPG/TRAIL ratio is a robust predictor of both short-term and long-term mortality in two independent cohorts after adjusting for comorbidities and cardiovascular risk factors[17]. Moreover, the OPG/TRAIL ratio provides important prognostic information for TA-AAD risk stratification.

The close link between a high OPG/TRAIL ratio and poor survival in TA-AAD may reflect the role of OPG and TRAIL in the development of aortic aneurysm/dissection. OPG, a cytokine of the TNF receptor superfamily, is expressed in vascular smooth muscle cells (VSMCs), endothelial cells (ECs), and osteoblasts. Various cytokines/hormones (i.e., TNF-α and angiotensin [Ang] II) promote the expression of OPG[18], which is increased in the medial layer of abdominal aortic aneurysm biopsy specimens[19]. OPG increases the expression of the Ang II type I receptor in aortic VSMCs and the expression of adhesion molecules in human ECs[18]. OPG deficiency was reported to inhibit aortic dilatation and rupture in apolipoprotein E-deficient mice[20,21]. Furthermore, recombinant OPG stimulates elastolytic activity in human aortic VSMCs[20]. TRAIL, a type II transmembrane protein also belonging to the TNF cytokine superfamily, is expressed by VSMCs, ECs, and immune

**Table 3 Performance of OPG/TRAIL ratio-guided risk stratification for predicting the risk of overall death in derivation and validation cohorts.**

| Cohort | Patient (%) | Risk Stratification | NPV | PPV | Sensitivity | Specificity |
|---|---|---|---|---|---|---|
| Derivation | 95 (17.7%) | Low risk | 95.0% | 23.4% | 94.5% | 20.8% |
| | 63 (11.8%) | High risk | 84.6% | 55.0% | 33.0% | 93.7% |
| Validation | 78 (19.5%) | Low risk | 96.2% | 26.4% | 96.6% | 24.0% |
| | 44 (11.0%) | High risk | 84.0% | 70.5% | 35.2% | 95.8% |

Thresholds developed from the derivation cohort were applied in validation cohort for verifying the performance of predicting overall death risk. *NPV* negative predictive value, *PPV* positive predictive value.

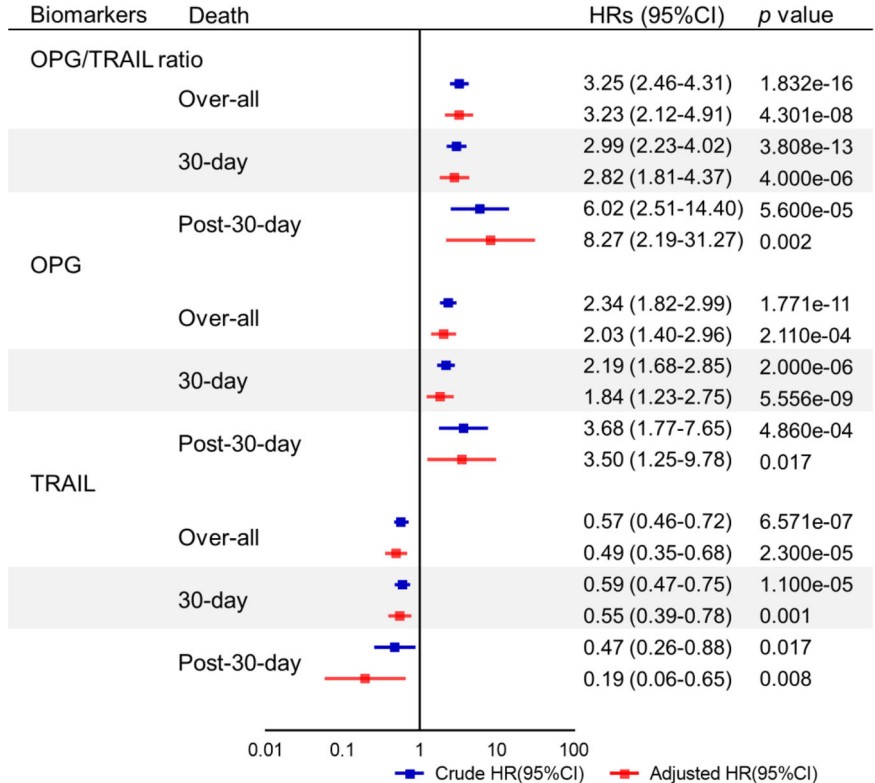

**Fig. 3 Cox proportional hazard analysis of OPG, TRAIL, and OPG/TRAIL ratio for predicting risk of death in validation cohort.** HRs with 95% CIs associated with 1-SD increase in OPG, TRAIL, and OPG/TRAIL ratio for overall death, 30-day death, and post-30-day death were plotted in validation cohort ($n = 400$). Models were adjusted for known risk factors (age $\geq$ 70 yrs, SBP, smoking), reported potential predictors (pain, malperfusion, shock or hypotension, coma or stroke), imaging indicators (thrombosis in false lumen and segment), and surgical treatment. Boxes represent HR. Error bars represent 95% CI. *p* values reported are two-tailed from COX proportional hazard regression analyses.

cells. TRAIL induces apoptosis of VSMCs and ECs by binding to the death receptor[22]. Expression of pro-apoptotic proteins is significantly increased in the aorta of patients with TA-AAD[23]. Plaque levels TRAIL are positively correlated with human plaque cell apoptosis and inflammatory activity[24]. The decrease in serum TRAIL levels in patients with TA-AAD might be due to overconsumption of TRAIL in the dissected tissue. In support of this hypothesis, TRAIL expression has been found to be increased in vulnerable plaques, but decreased in serum in patients with acute coronary syndromes[25]. Furthermore, OPG acts as a decoy receptor for TRAIL and a receptor activator of nuclear factor κB ligand. This phenomenon might be explained by the binding of OPG with TRAIL, which may inhibit the rapid clearance of TRAIL from the serum and stabilise its levels, thus augmenting its actions. Therefore, both OPG and TRAIL play important roles in multiple pathways contributing to aortic dissection, including apoptosis of VSMCs or ECs, inflammation, and degradation of the extracellular matrix.

In addition to the evidence provided by the animal models described above, clinical studies have also indicated the potential roles of OPG and TRAIL in aortic aneurysm. Koole et al. revealed that the concentration of OPG was positively correlated with the activity of matrix metalloproteinase (MMP)-2/9 in biopsies from patients with abdominal aortic aneurysm[19]. Moran et al. found that OPG was significantly associated with progression of abdominal aortic aneurysm in a cohort followed for 3 years[26]. The association between the OPG/TRAIL ratio and adverse prognosis in our study further strengthens the notion of a vital role of OPG and TRAIL in the pathogenesis of AAD, presenting the possible therapeutic potential of targeting OPG and TRAIL.

Reliable risk stratification of patients with TA-AAD remains a challenging task because of significant patient heterogeneity. Researchers are increasingly interested in using circulating biomarkers to improve risk stratification and clinical decision-making in several cardiovascular diseases[27]. Reclassification for

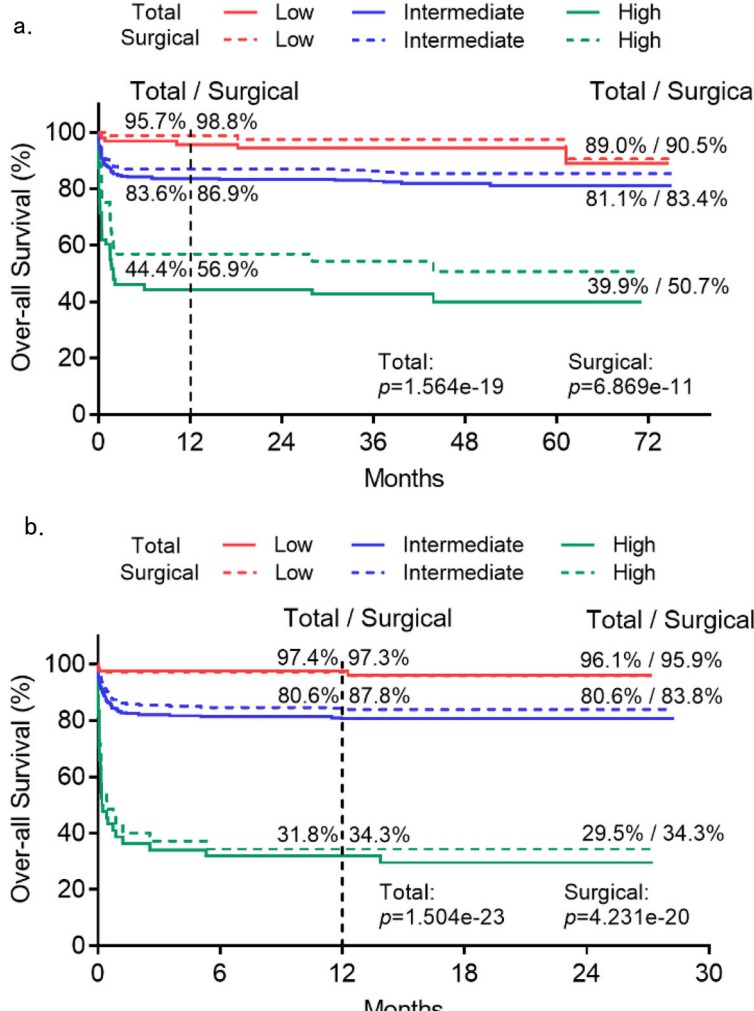

**Fig. 4 Kaplan–Meier curves for overall death according to the OPG/TRAIL ratio-guided risk stratum in the derivation and validation cohorts.** Overall survival percent (%) was plotted according to the OPG/TRAIL ratio-guided risk stratification in total and surgical patients in derivation cohort ($n = 536$) (**a**) and validation cohort ($n = 400$) (**b**). There were 462 (86.2%) and 363 (90.8%) patients who received surgical treatment in derivation and validation cohorts, respectively. $p$ values reported are two-tailed from log-rank tests.

mortality was significantly improved after adding the OPG/ TRAIL ratio to the existing AAD risk score or known clinical predictors. The present study findings support the use the OPG/ TRAIL ratio as an effective risk stratification tool, which has clinical value for the following reasons.

First, the optimal surgical strategy and postoperative medical therapy for patients with TA-AAD remain topics of debate. The preferred treatment option (total arch replacement (TAR) vs. hemiarch replacement) for patients with TA-AAD has yet to be determined. In patients with DeBakey type I TA-AAD (accounting for more than 70% of cases) with dissection extending beyond the ascending aorta, TAR is the conventional therapy for prevention of progressive dilation and development of aneurysm[28]. However, such an invasive surgery and its drawbacks (such as prolonged cardiopulmonary bypass and deep hypothermic circulatory arrest) present a formidable challenge that can lead to increased postoperative complications and death[29]. The risks of TAR in high-risk patients may outweigh its potential long-term benefits. Recent analyses from the IRAD show that patients who undergohemiarch replacement do not have poor long-term outcomes compared with TAR[30]. Compared with TAR, hemiarch replacement is relatively easy to perform. It is important to identify high-risk patients who could undergo hemiarch replacement, with a good chance of survival. The OPG/TRAIL ratio-

guided risk stratification may provide information on how to select patients who will benefit from TAR or hemiarch replacement. Second, serial radiographic monitoring and medical therapy are the main management issues during the follow-up phase. Extensive monitoring should be performed to evaluate late aortic complications (including progression of dissection, re-dissection, et al.) in high-risk patients. Thus, OPG/TRAIL ratio-guided risk stratification may have significant clinical and public health importance. However, the OPG/TRAIL ratio will never be the only variable determining such a far-reaching decision, which must also consider other aspects, such as the patient's preoperative comorbidities and intraoperative and postoperative situations.

The present study has several limitations. First, the surgical treatment provided was personalized at the discretion of each attending surgeon based on each patient's actual situation. However, the limited number of deaths did not allow us to stratify further (e.g., by operation type or operator). Second, the cut-off OPG/TRAIL ratio values for risk stratification should be interpreted cautiously because they were derived from a moderate sample size. However, validation in an independent cohort in the current study facilitated the clinical applicability of the results. Third, the present study did not include patients in special risk situations (e.g., TA-AAD in pregnancy). The prognostic value of the OPG/TRAIL ratio in such patients remains to be elucidated.

Moreover, since this cohort study was performed in an Asian population, additional studies are warranted to validate these findings in other ethnicities for generalizability. Fourth, there is a low number of subjects for the screening phase. Because protein profiling experiments are costly, the initial screening step is often performed in a small number of subjects. The small sample size might have led to an increase in false-positive and false-negative results. In this study, the validation of candidate biomarkers in a larger cohort eliminated false positives. However, we acknowledge that some biomarkers might be missed in underpowered studies.

In conclusion, this study demonstrates that the serum OPG/TRAIL ratio at admission is a robust prognostic marker of both short-term and long-term mortality in patients with TA-AAD, and will provide valuable information for risk stratification of these patients.

## Methods

**Study design**. This study had a multicentre, observational, prospective design and was performed at five hospitals of different tiers (national and provincial, non-overlapping geographic areas) in China. It included the following three stages: (i) a screening stage, during which candidate TA-AAD-specific proteins were identified using a 3-step screening process in the cross-sectional component of the study (120 TA-AAD vs. 244 healthy controls, recruited from Beijing Anzhen Hospital); (ii) a derivation stage, in which the prognostic value of candidate TA-AAD-specific proteins were prospectively evaluated and a prognostic biomarker-based risk stratification was derived in a multicentre cohort, which includes an internal dataset ($n = 300$, recruited from Beijing Anzhen Hospital) and an external dataset ($n = 236$, recruited from The First Affiliated Hospital of Sun Yat-sen Univerity, Xi Jing Hospital, The First Affiliated Hospital of Dalian Medical University, and Tongji Hospital) of patients with TA-AAD; and (iii) a validation stage, during which the predictive efficiency of the biomarker was verified in an independent cohort of patients with TA-AAD ($n = 400$). A flow chart of the study is shown in Fig. 5. The study is registered at ClinicalTrials.gov (NCT03010514) with the title "A Registry Study on Genetics and Biomarkers of Thoracic Aortic Aneurysm/

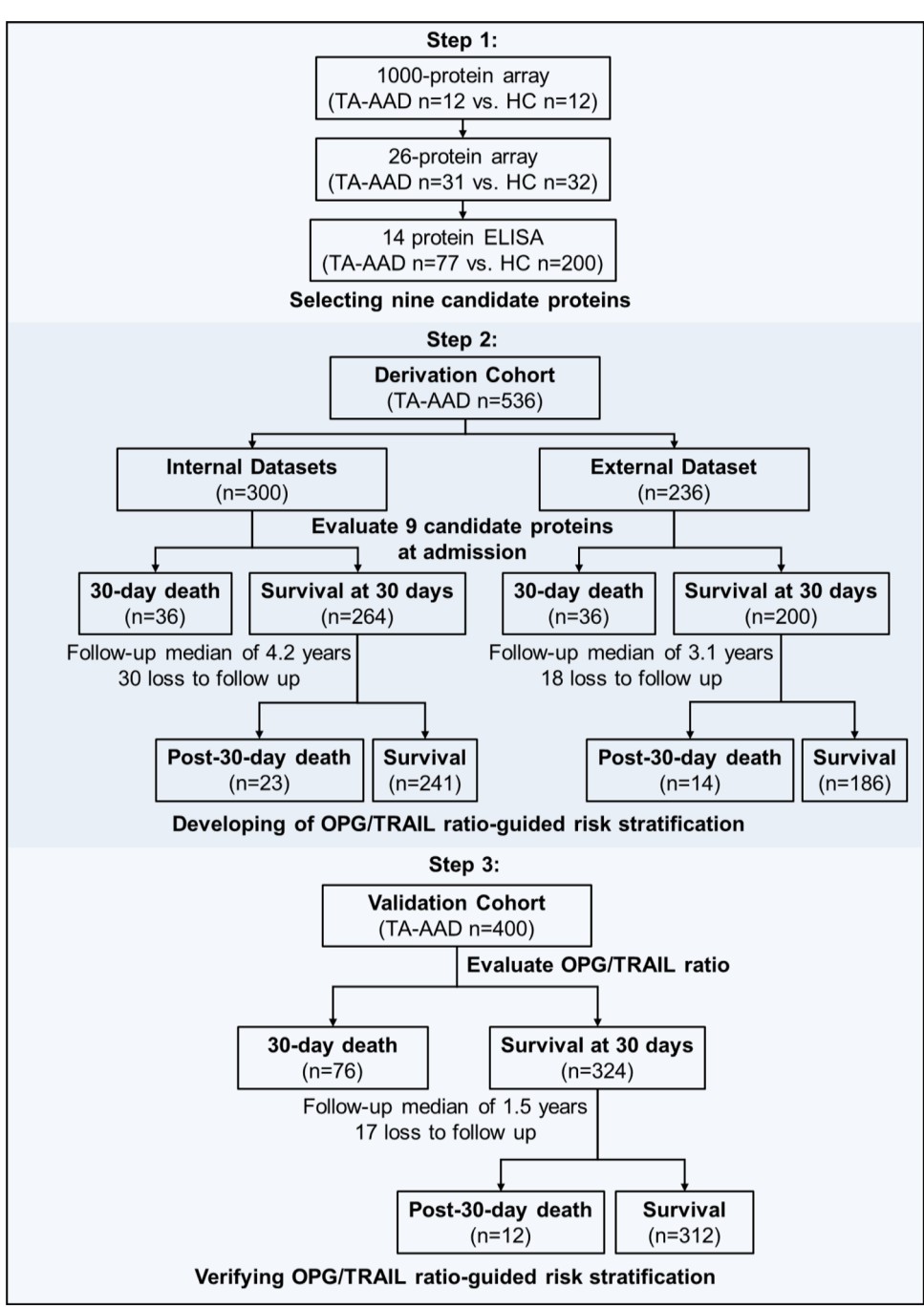

**Fig. 5 Study design.** HC healthy control, TA-AAD Type A acute aortic dissection.

Dissection". The study protocol could be accessed in the Supplementary Information.

**Study population**. The participants in the screening and an internal dataset of derivation cohort and a validation cohort were recruited from consecutive patients admitted to the Beijing Anzhen Hospital of Capital Medical University (Beijing, China). Between September 2014 and December 2016, 496 patients were admitted with a diagnosis of TA-AAD. Four hundred and twenty patients were enroled and 76 were excluded based on stringent eligibility criteria (Supplementary Fig. 7). The diagnosis of TA-AAD was confirmed based on patient history, imaging tests (computed tomography angiography or transthoracic echocardiography), and surgical findings. Healthy control subjects were recruited from individuals who underwent regular physical examinations at Beijing Anzhen Hospital, and their healthy status was further confirmed by transthoracic echocardiography to rule out TA-AAD, other thoracic aortic diseases, or abnormal cardiac structure/function. Using the same inclusion and exclusion criteria, a total of 236 patients with TA-AAD in the external dataset of derivation cohort were enroled between January 2017 and December 2017 from four other hospitals in China, namely, The First Affiliated Hospital of Sun Yatsen Univesity ($n = 46$, Guangzhou), Xi Jing Hospital ($n = 110$, Xian), The First Affiliated Hospital of Dalian Medical University ($n = 38$, Dalian), and Tongji Hospital ($n = 42$, Wuhan). The validation cohort included 400 consecutive patients with TA-AAD recruited between August 2018 and September 2019. All these hospitals are large university-based medical centres. Information of demographic characteristics, history, clinical presentation, physical examination, imaging information, and management were obtained from the medical records. All patients received standard surgical or medical treatments. A peripheral venous blood sample was drawn from all patients within 24 h of admission and before administration of treatment. This study was approved by the ethics committees or institutional review boards of all five medical centres. All study participants provided written informed consent. The study design and conduct complied with all relevant regulations regarding the use of human study participants and was conducted in accordance with the Declaration of Helsinki.

**Outcome measures**. Follow-up of all-cause mortality was initiated upon admission. The follow-up duration in both the derivation and validation cohorts ended at the death or termination time (October 2020). All patients enroled in the study were followed up for at least one year. The median follow-up time was 3.5 years [IQR 2.3–4.3] and 1.5 years [IQR1.0–1.9] in the derivation and validation cohorts, respectively. The median follow-up times were 4.2 years [IQR 2.9-4.9] years and 3.1 years [IQR 0.7–3.5] years in an internal and external datasets of derivation cohort. The primary endpoint was overall mortality and the secondary endpoints were 30-day and post-30-day mortality. Overall mortality was defined as all-cause mortality following initial hospital admission. Short-term mortality was defined as all-cause mortality within 30 days of admission. For post-30-day mortality, day 30 after admission was set as the start of the follow-up period. The survival status of each patient was confirmed by reviewing medical records and/or contacting each patient or their relatives individually. The causes of death in derivation and validation cohort are shown in Supplementary Table 9.

**Statistical analysis**. Continuous variables are summarised as the mean ± standard deviation and 95% CI and categorical variables as the number and percentage. Differences between any two groups were compared using a two-sample $t$-test (for normally distributed continuous variables, with log transformation as needed), the Mann–Whitney U test (for non-normally distributed continuous variables), or the Fisher's exact or chi-squared tests (for categorical data). The proportional hazard assumption was tested for each time-to-event outcome using the Schoenfeld residuals test, and no proportion hazard assumption was violated for the biomarker variables. Cox proportional hazards regression analysis was conducted to identify associations of candidate biomarkers with mortality. Known risk factors (age ≥ 70 years, high systolic blood pressure, and smoking)[15], reported potential predictors (pain, malperfusion, shock or hypotension, coma or stroke)[3], and imaging indicators (thrombosis in a false lumen and segments), and surgical treatment were considered as covariates. In the multivariate analysis, both predictors and covariates were included for each clinical outcome. Tests for multicollinearity revealed that the variance inflation factor was <2 for all input variables, suggesting that multicollinearity did not significantly affect the analysis. We performed multiple imputations of the missing data for the independent variables included in the regression model. The statistical models included biomarkers as continuous variables at a natural-log transformation scale and HRs were expressed per 1 standard derivation increase in the biomarker level. Possible nonlinear relationships between the OPG/TRAIL ratio and overall mortality were examined using restricted cubic splines. Analyses were multivariable-adjusted and used 3 knots, and the 5% highest and lowest biomarker observations were trimmed.

We examined the association between the OPG/TRAIL ratio and time to event in different subgroups (male vs. female, age ≥ 60 years vs. age < 60 years, smoking vs. non-smoking, hypertension vs. non-hypertension, diabetes vs. non-diabetes). This approach allowed us to estimate the subgroup-specific OPG/TRAIL ratio HR value and to compare the HRs in the two categories of the variables in the subgroup that differed. Discrimination was assessed by computing Harreller's C-index[31]. Changes in NRI[32] with addition of the OPG/TRAIL ratio were also calculated. We calculated the three-category NRI with the risk categories as <5% (low risk), 5–20% (medium risk), and >20% (high risk), which were chosen in accordance with the observed 1-year mortality of about 20% in the present study and the lowest mortality reported previously in patients after repair of TA-AAD[33] or two-category NRI with the risk categories as ≤20% and >20%. Three categories have three ways of moving up or down: low-medium; medium-high; and low-high. An event whose risk category changes from high risk to low risk is a more serious error than an event moving from high risk to medium risk. Thus, we mainly considered the three-category NRI with thresholds at 0.05 and 0.2 defining risk of low, medium, and high. The C-index and NRI were tested using 1000 bootstrap resamples. Cumulative death-free survival curves were derived using the Kaplan–Meier method and group-wise comparisons were based on the log-rank test.

Time-dependent receiver-operating characteristic curve analysis was performed to determine the cut-off value of the OPG/TRAIL ratio that predicted the overall risk of death. The criteria for threshold selection were as follows. First, a threshold false omission rate of ≤5% (meaning an NPV of 95%) was taken as an acceptable level of risk when categorising patients as "low risk" for overall mortality[34]. Conversely, a risk level of ≥50% for overall mortality (meaning a PPV of 50%) was used as the threshold to define "high risk" status[35]. Second, we further assessed the proportions of patients classified as low risk based on different NPVs (95%) or as high risk based on different PPVs (50%). The proportion of patients classified as low-risk or high-risk should never go below 10%. We assessed the quality of the cut-off value derived from the derivation cohort by calculating the sensitivity, specificity, PPV, and NPV in the external cohort.

The p-values were adjusted for multiple comparisons (number of proteins) using Benjamini and Hochberg correction for antibody array analysis[36]. All statistical analyses were performed using SPSS for windows (version 23.0; IBM Corp., Armonk, NY,USA) and R (version 3.3.3; R Foundation for Statistical Computing, Vienna, Austria). We used the rms package (version 6.2-0), survival package (version 3.2-10), and survcomp package (version 1.30.0) in R programming.

Sample size consideration, TA-AAD-associated definitions, methods used to collect the blood sample, details concerning the antibody array, bioinformatics analyses, and the enzyme-linked immunosorbent assay used are described in the Supplementary Information.

**Reporting summary**. Further information on research design is available in the Nature Research Reporting Summary linked to this article.

## Data availability
The raw data supporting the findings of the study have been provided in source data. Due to patient privacy protection purposes, the clinical data are not publicly available. Any individual affiliated with an academic institution may request access to the clinical data from corresponding authors (Yu.L. or J.D.) for research purposes. Data will be provided with a signed data access agreement (Supplementary Fig. 8). The timeframe for responding to an access to information is a 20-working day from the date of receipt. Source data are provided with this paper.

## Code availability
The R code used are available from the corresponding author upon request.

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

## Acknowledgements

We thank Dr. Jian Cui (Shanghai BioGenius Biotechnology Co., Ltd) for bioinformatics assistance. This work was supported by the National Science Foundation of China [grant numbers 91539121, 81930014, 81870339); Key Laboratory of Remodeling-Related Cardiovascular Diseases, Ministry of Education, China [grant number PXM2014-014226-000012); National Key R&D Program of China [grant number 2016YFC0903001].

## Author contributions

Yu.L., J.D., P.L., and J.Z. contributed to the design of the clinical cohorts. J.L., P.L., Ya.L., H.Y., W.D., J.O., Y.H., and L.W. contributed to collection of blood sample and clinical information. J. L. performed data analyses. H.Z. and X.P. contributed to statistical guidance and review. Yu.L. wrote the manuscript. Yu.L. and J.D. guided the study and approved the final version.

## Competing interests

The authors declare no competing interests.

## Additional information

**Peer review information** *Nature Communications* thanks Christoph Anton Nienaber and the other, anonymous reviewer(s) for their contirbution to the peer review of this work. Peer review reports are available.

