## [Peer Review File · Nature Communications]

REVIEWER COMMENTS

Reviewer #1 (Remarks to the Author):

This is an interesting manuscript with an important finding and those results merit publication. I believe however that the discussion is a bit long and could be shortened. Moreover, the bibliography is partially old and dated and thus should be updated (and those very old ones eliminated).

Reviewer #3 (Remarks to the Author):

1. The observed associations are strong but there is not reference for comparison with other markers. Why not show us the other seven? Or d-dimer?
2. It seems that the strengths of association are shown per 1 SD change in the biomarker. What is the distribution of OPG, TRAIL and the ratio?
3. The total number of events is small. Are there other centers/patients that could be added to this analysis?
4. Was follow-up for all-cause mortality initiated at admission and censored at 1-year or time of death? I think this analysis should be considered primary.
5. It is not clear what NRI was used. It would seem that NRI at event rate would be the most appropriate here.
6. P-values for change in c-index and NRI are not appropriate. Only effects with 95% CIs should be given. These CIs should also be added to event and non-event NRI components.
7. C-index for a coin toss would be 0.50. How can you get a value that is lower (i.e. 0.49?). This might signal an programming error.
8. Your criteria for threshold selection seem reasonable. You should show what % patients fall in each group in Table 3.
9. Your figures are not very informative. It would be nice to provide a spline figure investigating any non-linearity of association.

Reviewer #4 (Remarks to the Author):

In their manuscript "Identification and validation of OPG/TRAIL ratio as a novel biomarker to predict short-term and long-term mortality in patients with type A acute aortic dissection" Li et al. performed a study to identify and validate candidate biomarkers that have a value to predict risk of death and guide risk stratification in patients with TA-AAD.

Protein markers were measured in two cohort groups and the ratio of OPG/TRAIL was identified as the best marker to be associated with TA-AAD as well as to have a value for prediction of risk of mortality.

In general, the study is conducted according to state of the art approaches in biomarker research. However, in my view, a real external validation in an independent samples (and an independent analysis) is missing and should be included.

Comments:

1. please explain the number of patients included in total in more detail. In the supplement it is stated that the initial screening approach was performed on 12 case vs 12 control samples. This is a low number of subjects for a screening phase.
2. page 8: known risk factors are defined as age equal to or above 70 years. However, for subgroup analyses, age stratification was used as age below vs above 50 year. Please explain.
3. Please include a comparison of OPG/TRAIL ratio to Ddimer for measures of risk prediction. As Ddimer is an already established marker, the value of novel markers needs to be much better than

Ddimer to be of interest.

4. Study characteristics should be mentoined first in the results section.

REVIEWER COMMENTS

Reviewer #1 (Remarks to the Author):

This is an interesting manuscript with an important finding and those results merit publication. I believe however that the discussion is a bit long and could be shortened. Moreover, the bibliography is partially old and dated and thus should be updated (and those very old ones eliminated).

Re: As suggested by reviewer, we have shortened the discussion section, and added 5 references and eliminated 10 old references.

Reviewer #3 (Remarks to the Author):

1. The observed associations are strong but there is not reference for comparison with other markers. Why not show us the other seven? Or D-dimer?

Re: We greatly appreciate the reviewer's constructive suggestions. We conducted a series of analyses to compare the predictive/discriminative value of OPG, TRAIL and OPG/TRAIL ratio with that of seven candidate biomarkers and D-dimer.

First, Cox regression analysis showed that the OPG/TRAIL ratio was the strongest independent predictor of overall mortality (adjusted hazard ratio [HR] 2.04 [95% confidence interval [CI] 1.48-2.82]; 2.64 [95% CI 1.71-4.07]), 30-day mortality (adjusted HR 2.05 [95% CI 1.35-3.12]; 2.33 [95% CI 1.42-3.82]) and post-30-day mortality (adjusted HR 2.07 [95% CI 1.25-3.43]; 4.68 [95% CI 1.72-12.69]) in the internal and external datasets, respectively (*Fig. 1, New Supplementary Table 3*). Higher D-dimer concentrations were associated with increased 30-day mortality but was not with overall mortality, which is consistent with previous reports (*Added into Result section, Page 6, 20-21 and Page 7, Lines 1-5*).

Second, the Harrell's concordance index (C-index) for the OPG/TRAIL ratio was the highest for overall death (0.72 [95% CI 0.65-0.78]; 0.77 [95% CI 0.71-0.83], 30-day death (0.72 [95% CI 0.64-0.81]; 0.75 [95% CI 0.68-0.83]), and post-30-day death (0.72 [95% CI 0.61-0.83]; 0.82 [95% CI 0.71-0.92]) in the internal and external datasets, respectively (*New Fig. 2, New Supplementary Table 4*). These data demonstrate that the OPG/TRAIL ratio was the strongest predictor among the novel candidate biomarkers and D-dimer (*Added into Result section, Page 7, Lines 5-11*).

2. It seems that the strengths of association are shown per 1 SD change in the biomarker. What is the distribution of OPG, TRAIL and the ratio?

Re: OPG, TRAIL values and OPG/TRAIL ratio are not normally distributed according to Kolmogorov-Smirnov test. As the data were distributed in a nearly normal fashion

after log transformation, biomarkers were log-transformed and then classified as per-SD to perform Cox regression model.

3. The total number of events is small. Are there other centers/patients that could be added to this analysis?

Re: We completely agreed with the reviewer that more patients should be added to this study. The predictive value of OPG/TRAIL ratio were verified in an independent cohort of consecutive participants with TA-AAD (n=400). The whole study design was summarized as follows:

This study had a multicentre, observational, prospective design and was performed at five hospitals of different tiers (national and provincial, non-overlapping geographic areas) in China. It included the following three stages: i) a screening stage, during which candidate TA-AAD-specific proteins were identified using a 3-step screening process in the cross-sectional component of the study (120 TA-AAD vs. 244 healthy controls, recruited from Beijing Anzhen Hospital); ii) a derivation stage, in which the prognostic value of candidate TA-AAD-specific proteins were prospectively evaluated and a prognostic biomarker-based risk stratification was derived in a multicentre cohort, which include an internal dataset (n=300, recruited from Beijing Anzhen Hospital) and an external dataset (n=236, recruited from The First Affiliated Hospital of Sun Yat-sen University, Xi Jing Hospital, The First Affiliated Hospital of Dalian Medical University, and Tongji Hospital) of patients with TA-AAD; and iii) a validation stage, during which the predictive efficiency of the novel biomarker was verified in an independent cohort of patients with TA-AAD (n=400). A flow chart of the study is shown in *Fig. 5 (Added into Method section, Page 14, Lines 14-21; Page 15, Lines 1-5)*.

The baseline clinical characteristics of the validation cohort are summarized in *Table 1*. The 30-day and overall mortality rates were 19.0% and 22.0%, respectively. The 30-day mortality was higher in the validation cohort than in the derivation cohort because of the higher rupture rate during hospitalization in the validation cohort (9.0% vs. 6.5%). Patients who died were more likely to be female and older, and lower body

mass index, thrombosis in a false lumen and surgical treatment rate than those who survived (*Added into Result section, Page 8, Lines 15-20*).

We validated the predictive value of the OPG/TRAIL ratio for risk of death. The OPG/TRAIL ratio was an independent predictor of overall mortality (adjusted HR 3.23 [95% CI 2.12-4.91]), 30-day mortality (adjusted HR 1.84 [95% CI 2.82-4.37]), and post-30-day mortality (adjusted HR 8.27 [95% CI 2.19-31.27]; *Fig. 3*). For overall mortality, The OPG/TRAIL ratio significantly increased the C-index when added to the AAD score (C-index 0.21 [95% CI 0.13-0.29]) and clinical predictors (C-index 0.12 [95% CI 0.07-0.18]). Addition of the OPG/TRAIL ratio improved reclassification of the AAD score (NRI 0.73 [95% CI 0.68-0.77]; NRI_e 0.32 [95% CI 0.22-0.42]; NRI_{ne} 0.41 [95% CI 0.35-0.46]) or clinical predictors (NRI 0.50 [95% CI 0.45-0.55]; NRI_e 0.19 [95% CI 0.11-0.28]; NRI_{ne} 0.31 [95% CI 0.26-0.36]; *Table 2, Supplementary Fig. 3.c-d*). The discrimination/reclassification ability of OPG/TRAIL ratio remained to be well for 30-day death and post-30-day death (*Supplementary Table 5*) (*Added into Result section, Page 9, Lines 1-11*).

Subgroup analysis showed that the HRs for the OPG/TRAIL ratio were comparable across the various predefined subgroups (*Supplementary Fig. 6*) (*Added into Result section, Page 10, Lines 5-6*). Thus, the association of OPG/TRAIL ratio with short-term and long-term death was robust.

4. Was follow-up for all-cause mortality initiated at admission and censored at 1-year or time of death? I think this analysis should be considered primary.

Re: Follow-up of all-cause mortality was initiated upon admission. To comprehensively evaluate the mortality risk associated with candidate biomarkers, the follow-up time in derivation cohort was extended. The follow-up duration in both the derivation and validation cohorts ended at the death or termination time (October 2020). All patients enrolled in the study were followed up for at least one year. The median follow-up time was 3.5 years (IQR 2.3-4.3) and 1.5 years (IQR 1.0-1.9) in the derivation and validation cohorts, respectively. The median follow-up times were 4.2 years [IQR:2.9-4.9] years

and 3.1 years [IQR:0.7-3.5] years in an internal and external datasets of derivation cohort. (*Added into Method section, Page 16, Lines 10-15*).

As suggested by the reviewer, the primary endpoint was revised as overall all-cause mortality. In addition, the short-term endpoint was adjusted from in-hospital death to 30-day death for the following reasons: Both in-hospital and 30-day death are frequently used for assessing short-term mortality rate. In-hospital measure are relatively straightforward to derive, but critics of such measures cited the potential biases due to differences in facilities' lengths of stay and discharge practices. This is a multicentre study involving five hospitals distributed over five provinces. The difference of hospital stays is obvious in five hospitals (shortest: 5.8 days [IQR 1.1-9.0]; longest: 11 days [IQR 8.0-15.0]).

Use the measure of 30-day mortality allows for standardization of follow-up time. The OPG/TRAIL ratio was still a strong predictor of 30-day mortality (adjusted HR 2.05 [95% CI 1.35-3.12]; 2.33 [95% CI 1.42-3.82]) and post-30-day mortality (adjusted HR 2.07 [95% CI 1.25-3.43]; 4.68 [95% CI 1.72-12.69]) in the internal and external datasets, respectively (Fig. 1, Supplementary Table 3). In addition, C-index) for the OPG/TRAIL ratio was the highest for 30-day death (0.72 [95% CI 0.64-0.81]; 0.75 [95% CI 0.68-0.83]), and post-30-day death (0.72 [95% CI 0.61-0.83]; 0.82 [95% CI 0.71-0.92]) in the internal and external datasets, respectively (*Supplementary Table 4*) (*Added into Method section, Page 7, Lines 1-3 and 7-9*).

5. It is not clear what NRI was used. It would seem that NRI at event rate would be the most appropriate here.

Re: We used continuous NRI in the original version. As reviewer correctly pointed out, it is important that in placing more deaths into higher category. The categorical NRI clearly evaluated how the patients that were distributed into three mortality categories (low, medium, high) from an existing model to that of a proposed model with an additional risk factor.

We calculated the category NRI with the risk categories as <5% (low risk), 5%-20% (medium risk), and >20% (high risk), which were chosen in accordance with the

observed 1-year mortality of about 20% in the present study and the lowest mortality reported previously in patients after repair of TA-AAD (*Added into Method section, Page 18, Lines 7-10*).

Addition of the OPG/TRAIL ratio improved the reclassification of the AAD score (NRI 0.56 [95% CI 0.52-0.60]; NRI_e 0.18 [95% CI 0.11-0.26]; NRI_{ne} 0.38 [95% CI 0.33-0.42]) or clinical predictors (NRI 0.23 [95% CI 0.20-0.27]; NRI_e 0.06 [95% CI 0.01-0.10]; NRI_{ne} 0.18 [95% CI 0.14-0.21]; *Table 2*). Details of reclassification improvements of OPG/TRAIL ratio on the existing models were shown in Supplementary Fig. 3a-b. There was a similar NRI pattern for the OPG/TRAIL ratio regarding 30-day and post-30-day deaths in patients with and without events (*Supplementary Table 5*) (*Added into Result section, Page 8, Lines 1-5*).

6. P-values for change in C-index and NRI are not appropriate. Only effects with 95% CIs should be given. These CIs should also be added to event and non-event NRI components.

Re: As suggested by reviewer, we only kept the 95% CIs for Δ C-index and NRI, and we added the 95% CIs of NRIs for event and non-event (*Table 2*).

7. C-index for a coin toss would be 0.50. How can you get a value that is lower (i.e. 0.49?). This might signal an programming error.

Re: We are apologized for this carelessness. A value of C-index (0.49) is a clerical error. Basal C-index of AAD score for 30-day death prediction should be 0.58 (95% CI 0.49-0.68). We have corrected it (*Supplementary Table 5*).

8. Your criteria for threshold selection seem reasonable. You should show what % patients fall in each group in Table 3.

Re: We appreciated the reviewer's suggestion. We have added the % patients in low-risk (17.7%, 19.5%) and high-risk (11.8%, 11.0%) group in the derivation cohort and validation cohort, respectively (*Table 3*).

Because of adding an independent validation cohort, we have recalculated the cut-off value of OPG/TRAIL in the whole derivation cohort (including internal and external datasets) for risk stratification. The criteria for threshold selection were as follows. First, a threshold false omission rate of $\leq 5\%$ (meaning an NPV of 95%) was taken as an acceptable level of risk when categorising patients as “low risk” for overall mortality. Conversely, a risk level of $\geq 50\%$ for overall mortality (meaning a PPV of 50%) was used as the threshold to define “high risk” status. Second, we further assessed the proportions of patients classified as low risk based on different NPVs (95%) or as high risk based on different PPVs (50%). The proportion of patients classified as low-risk or high-risk should never go below 10% (*Added into Method section, Page 18, Lines 14-20*).

Based on the criteria for threshold selection, two cut-off OPG/TRAIL ratio values (4 and 33) were chosen for categorising 95 patients (17.7%) into low-risk (<4) yielding NPV of 95% and 63 patients (11.8%) into high-risk (>33) yielding PPV of 55% (*Table 3, Supplementary Fig. 4*) (*Added into result section, Page 8, Lines 8-11*). This cut-off value was same with original threshold derived from the internal datasets.

9. Your figures are not very informative. It would be nice to provide a spline figure investigating any non-linearity of association.

Re: As suggested by reviewer, possible nonlinear relationships between the OPG/TRAIL ratio and overall mortality were examined using restricted cubic splines. Analyses were multivariable-adjusted and used 3 knots, and the 5% highest and lowest biomarker observations were trimmed (*Added into Method section, Page 17, Lines 18-21*).

Multivariable-adjusted restricted cubic spline analyses of the association between the OPG/TRAIL ratio and overall mortality provided no evidence of a nonlinear association ($p=0.244$; 0.231) and indicated a significant linear association ($p< 0.001$) in the derivation and validation cohorts (*Supplementary Fig. 5*) (*Added into Result section, Page 10, Lines 2-5*).

Reviewer #4 (Remarks to the Author):

In their manuscript "Identification and validation of OPG/TRAIL ratio as a novel biomarker to predict short-term and long-term mortality in patients with type A acute aortic dissection" Li et al. performed a study to identify and validate candidate biomarkers that have a value to predict risk of death and guide risk stratification in patients with TA-AAD. Protein markers were measured in two cohort groups and the ratio of OPG/TRAIL was identified as the best marker to be associated with TA-AAD as well as to have a value for prediction of risk of mortality. In general, the study is conducted according to state-of-the-art approaches in biomarker research. However, in my view, a real external validation in an independent sample (and an independent analysis) is missing and should be included.

Re: We appreciated the Reviewer's constructive suggestion. We performed the real validation in an independent cohort. The validation cohort had no participant overlap with the derivation cohort. We evaluated the prediction/discrimination/reclassification ability of OPG/TRAIL ratio with the three outcomes. Please see the Response 3 to Reviewer 3 about the results of Cox progression, C-index and NRI analyses.

In addition, we evaluated the performance of OPG/TRAIL-based risk stratification in the validation cohort. According to the cut-off values for the OPG/TRAIL ratio derived from the derivation cohort, the cut-off value of <4 had an NPV of 96.2% and a sensitivity of 96.6% whereas the cut-off value >33 had a PPV of 70.5% and a specificity of 95.8% for predicting the overall risk of mortality. Moreover, the cut-off OPG/TRAIL ratio values also showed a better NPV (97.4%) and PPV (61.4%) for 30-day mortality in the validation cohort (*Table 2, Supplementary Table 6*). The PPV value of 23.5% was low for post-30-day mortality due to the low number of deaths ($n=4$) (*Added into Result section, Page 9, Lines 12-18*).

Survival analyses showed that patients at high risk in the validation cohorts had a lower 1-year overall survival rate than patients at low risk (31.8% vs. 97.4%), even in

the patients that underwent surgery (34.3% vs. 97.3%; Fig. 4) (*Added into Result section, Page 9, Lines 18-21*).

Comments:

1. please explain the number of patients included in total in more detail. In the supplement, it is stated that the initial screening approach was performed on 12 case vs 12 control samples. This is a low number of subjects for a screening phase.

Re: We explained the considerations of sample size in different stages. In derivation cohort, to achieve 90% power at 5% significance with a 4:1 sampling ratio (because 25% of reported mortality rate), we required 504 participants to detect an overall survival hazard ratio of 1.5. In validation cohort, the sample size estimation based on preliminary data of derivation cohort. Assuming a mortality rate of 20%, a sample size of 187 patients would be powered (90% at $p < 0.05$) to detect a HR of OPG/TRAIL ratio of 2.0 (*Added into Supplemental Method section, Page3, Lines2-8*).

Because protein profiling experiments are costly, the initial screening step is often performed in a small number of subjects. The small sample size might have led to an increase in false-positive and false-negative results. In this study, the validation of candidate biomarkers in a larger cohort eliminated false positives. However, we acknowledge that some biomarkers might be missed in underpowered studies (*Added into the limitation of Discussion section, Page 14, Lines 2-7*).

2. page 8: known risk factors are defined as age equal to or above 70 years. However, for subgroup analyses, age stratification was used as age below vs above 50 year. Please explain.

Re: Old age (≥ 70 years) is a known risk factor for patients with TA-AAD, supported by high mortality of 47.6% in the derivation cohort. However, the sample size of patients aged over 70 (only 21 patients) is too small for Cox regression analyses. We appreciated the reviewer's question and rethink the reason for subgroup stratified by age. We calculated the mortality in different ages by decade. The mortality rate began to rise significantly at 60 years old in the derivation cohort (18.3% [<60 years] vs. 30.7%

[≥ 60 years], $p=0.008$) and validation cohort (16.9% [<60 years] vs. 39.1% [≥ 60 years], $p<0.001$). Thus, we reanalyzed the HRs in subgroups according to age above vs. below 60 years. OPG/TRAIL ratio remained associated with overall mortality across age subgroups (*Supplementary Fig. 6*).

3. Please include a comparison of OPG/TRAIL ratio to D-dimer for measures of risk prediction. As D-dimer is an already established marker, the value of novel markers needs to be much better than D-dimer to be of interest.

Re: We agreed with the opinion that a comparison of OPG/TRAIL ratio to D-dimer for measures of risk prediction should be included.

Higher D-dimer concentration was associated with increased 30-day mortality (adjusted HR 1.54 [95% CI 1.03-2.30]; 1.73, [95% CI 1.11-2.70]) but was not related to post-30-day mortality (adjusted HR 0.92 [95% CI 0.54-1.54]; 0.49 [95% CI 0.21-1.12]) in the internal and external datasets, respectively (*Supplementary Table 3*).

C-index illustrated that OPG/TRAIL ratio (basal C-index 0.72 [95% CI 0.65-0.78]; 0.77 [95% CI 0.71-0.83]) was superior to D-dimer (basal C-index 0.56 [95% CI 0.47-0.64]; 0.54 [95% CI 0.47-0.62]) in the internal and external datasets, respectively. For 30-day and post-30-day death, OPG/TRAIL ratio remained superior to D-dimer (*Fig. 2, Supplementary Table 4*).

4. Study characteristics should be mentioned first in the results section.

Re: As suggested by reviewer, we have moved the “Characteristics of study population” to the first part of result section.

REVIEWERS' COMMENTS

Reviewer #1 (Remarks to the Author):

This reviewer is happy to accept the current version of the manuscript for publication as is. The open resolvable issues have been addressed satisfactorily. An editorial commentary may address certain aspects that could not be fully resolved to everybody's satisfaction.

Reviewer #3 (Remarks to the Author):

The authors have done a good job responding to the comments and the paper has been greatly improved. Two outstanding comments:

1. Given the focus on the 1-year event rate (please make it more explicit throughout) and the fact that this rate is about 20%, NRI at 20% would be the best. Adding the 5-20% category seems to unnecessarily inflate the value of the NRI.
2. Please give references to Harrell's c statistic and Pencina's NRI so it is clear what was used.

Reviewer #4 (Remarks to the Author):

Thank you for replying to my comments. In my view all comments have been adequately replied to.

Although the low sample size in the first step of your approach still concerns me, you have adequately stated this in the manuscript.

One additional suggestion is, since the manuscript is based on an Asian population you should include a statement that your finding should be validated in other ethnicities for generalizability.

REVIEWER COMMENTS AND RESPONSES

Reviewer #3 (Remarks to the Author):

The authors have done a good job responding to the comments and the paper has been greatly improved. Two outstanding comments:

1. Given the focus on the 1-year event rate (please make it more explicit throughout) and the fact that this rate is about 20%, NRI at 20% would be the best. Adding the 5-20% category seems to unnecessarily inflate the value of the NRI.

Re: We appreciated the Reviewer's constructive suggestion. We calculated the category NRI with the risk categories as $\leq 20\%$ and $>20\%$ as following Table. For overall death, when added to the AAD risk score or clinical predictors, NRI for OPG/TRAIL ratio improved as a result of reclassification of both participants who died or did not die in two cohorts. For 30-day death and post-30-day death, NRI for OPG/TRAIL ratio was dominated by an improved classification of subjects who died, and had little change in classification of subjects who survived. It is important placing more deaths into higher category.

Table. NRI for OPG/TRAIL ratio over AAD score or clinical predictors in derivation and validation cohorts.

	Derivation Cohort			Validation Cohort		
	Over-all death NRI (95%CI)	30-day death NRI (95%CI)	Post-30-day death NRI (95%CI)	Over-all death NRI (95%CI)	30-day death NRI (95%CI)	Post-30-day death NRI (95%CI)
AAD score	0.53(0.49-0.57)	-0.01(-0.02- 0.00)		0.64(0.60-0.69)	0.34(0.29-0.38)	
+O/T ratio	Events:	Events:		Events:	Events:	
	0.21(0.13-0.29)	0.13(0.05-0.20)		0.31(0.21-0.41)	0.33(0.22-0.44)	
	No-events:	No-events:		No-events:	No-events:	
	0.32(0.28-0.37)	-0.14(-0.17- -0.11)		0.34(0.28-0.39)	0.01(0.00-0.02)	
Clinical Predictors	0.17(0.14-0.21)		0.05(0.03-0.07)	0.29(0.25-0.34)		0.24(0.19-0.29)
+O/T ratio	Events:		Events:	Events:		Events:
	0.09(0.04-0.15)		0.09(0.00-0.18)	0.17(0.09-0.25)		0.25(-0.04-0.54)
	No-events:		No-events:	No-events:		No-events:
	0.08(0.06-0.11)		-0.04(-0.05- -0.02)	0.12(0.09-0.16)		-0.01(-0.02-0.00)

Clinical predictors included stroke, chronic renal dysfunction, myocardial infarction, older age, female, atherosclerosis history and previous cardiac surgery. O/T=OPG/TRAIL.

The purpose of risk stratification is to guide appropriate treatment decisions. When categories correspond to treatment decisions, reclassification is not just the direction. Three-categories have three ways of moving up or down: low-medium; medium-high; and low-high. An event whose risk category changes from high risk to low risk is a more serious error than an event moving from high risk to medium risk. Thus, we considered the three-categories NRIs with thresholds at 0.05 and 0.2 defining risk of low, medium and high.

2. Please give references to Harrell's c statistic and Pencina's NRI so it is clear what was used.

Re: As suggested by Reviewer, we have added the references to Harrell's c statistic (*Overall C as a measure of discrimination in survival analysis: model specific population value and confidence interval estimation. Stat Med. 2004;23(13), 2109-2123.*) and Pencina's NRI (*Evaluating the added predictive ability of a new marker: from area under the ROC curve to reclassification and beyond. Stat Med. 2008;27(2),157-172; discussion 207-212.*) in the Methods (Page 18, Line 11).

Reviewer #4 (Remarks to the Author):

Thank you for replying to my comments. in my view all comments have been adequately replied to. Although the low samples size in the first step of your approach still concerns me, you have adequately stated this in the manuscript.

One additional suggestion is, since the manuscript is based on an Asian population you should include a statement that your finding should be validated in other ethnicities for generalizability.

Re: We appreciated the Reviewer's constructive suggestion. The sentence "Since this cohort study was performed in an Asian population, additional studies are warranted to validate these findings in other ethnicities for generalizability." has been added into the limitation of Discussion section (*Page 14, Lines 2-4*).